# Systematic Weight Pruning of DNNs using Alternating Direction Method of Multipliers

**Tianyun Zhang, Shaokai Ye, Yipeng Zhang, Yanzhi Wang & Makan Fardad**
Department of Electrical Engineering and Computer Science
Syracuse University,
Syracuse, NY 13244, USA
`{tzhan120,sye106,yzhan139,ywang393,makan}@syr.edu`

## Abstract

We present a systematic weight pruning framework of deep neural networks (DNNs) using the alternating direction method of multipliers (ADMM). We first formulate the weight pruning problem of DNNs as a constrained nonconvex optimization problem, and then adopt the ADMM framework for systematic weight pruning. We show that ADMM is highly suitable for weight pruning due to the computational efficiency it offers. We achieve a much higher compression ratio compared with prior work while maintaining the same test accuracy, together with a faster convergence rate.

## 1 Introduction

Despite the significant achievements enabled by DNNs, their large model size and computational requirements will add a significant burden to state-of-the-art computing systems (Krizhevsky et al., 2012; Simonyan & Zisserman, 2014; Han et al., 2016), especially for embedded and IoT systems. As a result, a number of prior works are dedicated to *weight pruning methods* in order to simultaneously reduce the computation and model storage requirements of DNNs, with minor effect on the overall accuracy.

A simple but effective method has been proposed in Han et al. (2015), which prunes the relatively less important weights and performs retraining for maintaining accuracy in an iterative manner. This method has been extended and generalized in multiple directions, including energy efficiency-aware pruning (Yang et al., 2016), structure-preserved pruning using regularization methods (Wen et al., 2016), and employing heuristics motivated by VLSI CAD (Dai et al., 2017). While existing pruning methods achieve good model compression ratios, they are heuristic, lack theoretical guarantees on compression performance, and require time-consuming iterative retraining processes.

To mitigate these shortcomings, we present a systematic framework of model compression, by (i) formulating the weight pruning problem as a constrained nonconvex optimization problem, and (ii) adopting the *alternating direction method of multipliers* (ADMM) (Boyd et al., 2011) for systematic weight pruning. Upon convergence of ADMM, we remove the weights which are (close to) zero and retrain the network. Our extensive numerical experiments indicate that ADMM works very well in practice and is highly suitable for weight pruning. Overall, we achieve a model that has much fewer weights and less computation than previous weight pruning work while maintaining the same test accuracy as the model before pruning. The proposed method has a faster convergence rate compared with prior works.

## 2 Problem Formulation and Proposed Framework

Consider an $N$-layer DNN, where the collection of weights in the $i$-th layer is denoted by $\mathbf{W}_i$. In a convolutional layer the weights are organized in a four-dimension tensor and in a fully-connected layer they are organized in a two-dimension matrix (Leng et al., 2017). The loss function associated with the DNN is represented by $f(\mathbf{W}_1, \ldots, \mathbf{W}_N)$. In this paper, our objective is to prune the weights of the DNN and therefore we minimize the loss function subject to constraints on the

cardinality of weights in each layer. Thus, our training process solves

$$\underset{\{\mathbf{W}_i\}}{\text{minimize}} \quad f(\mathbf{W}_1, \dots, \mathbf{W}_N), \quad \text{subject to} \quad \mathbf{W}_i \in \mathbf{S}_i = \{\mathbf{W} \mid \mathrm{card}(\mathbf{W}) \le l_i\}, \; i = 1, \dots, N,$$

(1)

where $\mathrm{card}(\cdot)$ returns the number of nonzero elements of its matrix argument and $l_i$ is the desired number of weights in the $i$-th layer of the DNN after pruning. It is clear that $\mathbf{S}_1, \dots, \mathbf{S}_N$ are non-convex sets, and it is in general difficult to solve optimization problems with nonconvex constraints. A recent paper of Boyd et al. (2011), however, demonstrates that ADMM can be utilized to solve nonconvex optimization problems in some special formats. The above problem can be equivalently rewritten in ADMM form as

$$\underset{\{\mathbf{W}_i\}}{\text{minimize}} \quad f(\mathbf{W}_1, \dots, \mathbf{W}_N) + \sum_{i=1}^{N} g_i(\mathbf{Z}_i), \quad \text{subject to} \quad \mathbf{W}_i = \mathbf{Z}_i,$$

where $g_i(\cdot)$ is the indicator function of $\mathbf{S}_i$

$$g_i(\mathbf{Z}_i) = \begin{cases} 0 & \text{if } \mathrm{card}(\mathbf{Z}_i) \le l_i, \\ +\infty & \text{otherwise.} \end{cases}$$

The augmented Lagrangian (Boyd et al., 2011) of the optimization problem is given by

$$L_\rho(\{\mathbf{W}_i\}, \{\mathbf{Z}_i\}, \{\mathbf{\Lambda}_i\}) = f(\mathbf{W}_1, \dots, \mathbf{W}_N) + \sum_{i=1}^{N} g_i(\mathbf{Z}_i) + \sum_{i=1}^{N} \mathrm{tr}\left[\mathbf{\Lambda}_i(\mathbf{W}_i - \mathbf{Z}_i)\right] + \sum_{i=1}^{N} \frac{\rho_i}{2} \|\mathbf{W}_i - \mathbf{Z}_i\|_F^2,$$

where the matrices $\{\mathbf{\Lambda}_1, \dots, \mathbf{\Lambda}_N\}$ are Lagrange multipliers, the positive scalars $\{\rho_1, \dots, \rho_N\}$ are penalty parameters, $\mathrm{tr}(\cdot)$ denotes the trace, and $\|\cdot\|_F^2$ denotes the Frobenius norm. With the scaled dual variable $\mathbf{U}_i = (1/\rho_i)\mathbf{\Lambda}_i$ the augmented Lagrangian can be equivalently expressed as

$$L_\rho(\{\mathbf{W}_i\}, \{\mathbf{Z}_i\}, \{\mathbf{U}_i\}) = f(\mathbf{W}_1, \dots, \mathbf{W}_N) + \sum_{i=1}^{N} g_i(\mathbf{Z}_i) + \sum_{i=1}^{N} \frac{\rho_i}{2} \|\mathbf{W}_i - \mathbf{Z}_i + \mathbf{U}_i\|_F^2 - \sum_{i=1}^{N} \frac{\rho_i}{2} \|\mathbf{U}_i\|_F^2.$$

The ADMM algorithm proceeds by repeating, for k = 0,1, . . ., the following steps (Boyd et al., 2011; Liu et al., 2013):

$$\{\mathbf{W}_i^{k+1}\} := \underset{\{\mathbf{W}_i\}}{\arg\min} \quad L_\rho(\{\mathbf{W}_i\}, \{\mathbf{Z}_i^k\}, \{\mathbf{U}_i^k\})$$

(2)

$$\{\mathbf{Z}_i^{k+1}\} := \underset{\{\mathbf{Z}_i\}}{\arg\min} \quad L_\rho(\{\mathbf{W}_i^{k+1}\}, \{\mathbf{Z}_i\}, \{\mathbf{U}_i^k\})$$

(3)

$$\mathbf{U}_i^{k+1} := \mathbf{U}_i^k + \mathbf{W}_i^{k+1} - \mathbf{Z}_i^{k+1},$$

(4)

until both of the following conditions are satisfied

$$\left\|\mathbf{W}_i^{k+1} - \mathbf{Z}_i^{k+1}\right\|_F^2 \le \epsilon_i, \quad \left\|\mathbf{Z}_i^{k+1} - \mathbf{Z}_i^k\right\|_F^2 \le \epsilon_i.$$

Problems (2) simplifies to

$$\underset{\{\mathbf{W}_i\}}{\text{minimize}} \quad f(\mathbf{W}_1, \dots, \mathbf{W}_N) + \sum_{i=1}^{N} \frac{\rho_i}{2} \left\|\mathbf{W}_i - \mathbf{Z}_i^k + \mathbf{U}_i^k\right\|_F^2,$$

where the first term is the loss function of the DNN, and the second term can be considered as a special $L_2$ regularization. Since the regularizer is a quadratic norm the complexity of minimizing the above loss function (for example, via gradient descent) is the same as the complexity of solving $\underset{\{\mathbf{W}_i\}}{\text{minimize}} \; f(\mathbf{W}_1, \dots, \mathbf{W}_N)$. On the other hand, problem (3) simplifies to

$$\underset{\{\mathbf{Z}_i\}}{\text{minimize}} \quad \sum_{i=1}^{N} g_i(\mathbf{Z}_i) + \sum_{i=1}^{N} \frac{\rho_i}{2} \left\|\mathbf{W}_i^{k+1} - \mathbf{Z}_i + \mathbf{U}_i^k\right\|_F^2.$$

Since $g_i(\cdot)$ is the indicator function of $\mathbf{S}_i$ the solution of this problem is explicitly found to be (Boyd et al., 2011)

$$\mathbf{Z}_i^{k+1} = \prod_{\mathbf{S}_i}(\mathbf{W}_i^{k+1} + \mathbf{U}_i^k),$$

(5)

Table 1: Weights pruning result on Lenet-300-100 network(without incurring accuracy loss)

| Layer | Weights | Weights after prune | Percentage of weights after prune |
|-------|---------|---------------------|-----------------------------------|
| fc1 | 784×300=235.2k | 11.76k | 5% |
| fc2 | 300×100=30k | 2.1k | 7% |
| fc3 | 100×10=1k | 0.12k | 12% |
| Total | 266.2k | 13.98k | 5.25% |

Table 2: Weights pruning result on Lenet-5 network(without incurring accuracy loss)

| Layer | Weights | Weights after prune | Percentage of weights after prune |
|-------|---------|---------------------|-----------------------------------|
| conv1 | 5×5×1×20=0.5k | 0.1k | 20% |
| conv2 | 5×5×20×50=25k | 2.5k | 10% |
| fc1 | 800×500=400k | 20k | 5% |
| fc2 | 500×10=5k | 0.35k | 7% |
| Total | 430.5k | 22.95k | 5.33% |

where $\prod_{\mathbf{S}_i}(\cdot)$ denotes Euclidean projection onto the set $\mathbf{S}_i$. Note that $\mathbf{S}_i$ is a nonconvex set, and computing the projection onto a nonconvex set is a difficult problem in general. However, the special structure of $\mathbf{S}_i = \{\mathbf{W} \mid \mathrm{card}(\mathbf{W}) \leq l_i\}$ allows us to express this Euclidean projection analytically. Namely, the solution of (5) is to keep the $l_i$ elements of $\mathbf{W}_i^{k+1} + \mathbf{U}_i^k$ with the largest magnitudes and set the rest to zero (Boyd et al., 2011). Finally, we update the dual variable $\mathbf{U}_i$ according to (4). This constitutes one iteration of the ADMM algorithm.

We observe that the proposed framework exhibits multiple major advantages in comparison with the heuristic weight pruning method of Han et al. (2015). Our proposed method achieves (i) higher convergence speed compared to iterative retraining, and (ii) higher compression ratio, as we demonstrate next.

## 3 EXPERIMENTS

We implement the network pruning method in Tensorflow (Abadi et al., 2016). We have tested weight pruning on the MNIST benchmark using the LeNet-300-100 and LeNet-5 model (LeCun et al., 1998). In our experiments on these networks, our proposed ADMM method converges in approximately 10 iterations. As mentioned before, problem (2) can be solved by gradient descent. In experiments, we found that the number of steps we need for solving problem (2) by gradient descent is approximately 1/5 of the number of steps for training the original network. On the other hand, problem (3) and (4) are straightforward to carry out, thus their computational time can be ignored. Therefore, the total computation time of the ADMM algorithm is approximately equal to training the original network twice. While the solution of problem (1) should render $W_i$ with only $l_i$ nonzero elements, the solution we obtain through ADMM contains additional small nonzero entries. To deal with this issue, we keep the $l_i$ largest magnitude elements of $W_i$, set the rest to zero and no longer involve these elements in training (i.e., we prune these weights). We then retrain the network with 1/10 of its original learning rate (Han et al., 2015).

Table 1 shows that our pruning reduces the number of weights by 19× on Lenet-300-100. Table 2 shows that our pruning reduces the number of weights by 18× on Lenet-5. Our pruning will not incur accuracy loss and can achieve a much higher compression ratio on these networks compared with the work of Han et al. (2015), which reduces parameters 12× on both networks. Furthermore, on Lenet-5 we can reduce the number of weights by 10× in convolutional layers, which is also higher than the 8× in the work of Han et al. (2015). Although on Lenet-5 the number of weights in convolutional layers is less than fully connected layers, the computation on Lenet-5 is dominated by its convolutional layers. This means that our pruning can reduce more computation compared with prior work.

ACKNOWLEDGMENTS

Financial support from the National Science Foundation under award ECCS-1609916 is gratefully acknowledged.

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
