# OpenReview forum: "Systematic Weight Pruning of DNNs using Alternating Direction Method of Multipliers"
_ICLR.cc/2018/Workshop — Accept_

### Official Review · AnonReviewer3 · 2018-03-11
**Intersting empirical study**

**Rating:** 6
**Confidence:** 3

**Review:**

This paper uses ADMM to justify hard thresholding heuristic, which thresholds out DNN's weights if the value is less than some amount.
I think the empirical case study is well carried, and recommend this for the workshop to have a discussion. Note that the convergence of non-convex ADMM is unknown, and this paper does not use ADMM in a strict sense. ADMM splits the optimization problem to multiple instances, and we need to solve those subproblems EXACTLY. This paper worked around this by obtaining very crude approximation of solution for the subproblem. But for practical purposes, I think this maybe enough.

Also I would encourage the authors to report training AND TESTING loss and accuracy. Non-convex ADMM has stability issue, so it's best to see

1. Run the algorithm 10 times.
2. Report error bar in training and testing loss and accuracy.

In the optimization point of view, reducing the number of non-zero weights while preserving training loss makes sense, but in the machine learning point of view, if the procedure hurts the generalization property significantly, it is not useful.
So I recommend to the authors to include a performance of weight-pruned classifier on the test set.

---

### Official Review · AnonReviewer1 · 2018-03-11
**A decent work, but with a few minor issues and unexplained aspects**

**Rating:** 6
**Confidence:** 5

**Review:**

This paper proposes to use an l0 norm regularizer (although the authors do not identify it as such) to achieve weight sparsity in DNN training. The authors propose to upper bound the l0 norm (equivalently, the number of non-zero components) of the weights of each layer, by adding an indicator function to the underlying loss function. The resulting (non-convex) optimization problem is then tackled via ADMM. The experimental results (though not very extensive) show that the approach is able to yield networks with many zero weights.

The paper suffers from a few minor issues and unexplained aspects. How is the update with respect to W_i carried out? Is it solved exactly, or simply a few steps of SGD? The notation in Equation (5) is wrong since it is the notation used to represent products; the authors should use \Pi rather than \prod. Furthermore, notice that this projection may not be unique, due to the non-convexity of S_i. How are the \rho_i parameters selected? Does this choice affect the results?

---

### Public Comment · ~Christian_Gagné1 · 2018-04-19
**Similar paper submitted at ICLR 2017 and on arXiv**

A very similar work was submitted at ICLR 2017 by my team.

https://openreview.net/forum?id=rye9LT8cee
https://arxiv.org/abs/1611.01590

It is a pity to see that it was not properly cited.

---

> ### Public Comment · ~Tianyun_Zhang1 · 2018-04-24
> **We have added a citation in the Arxiv version**
>
> Thanks for the comment. We were unaware of this manuscript during our submission. Now we have added a citation to yours in the Arxiv version.
>
>
> These two papers have different focuses. Your submission uses ADMM to solve DNN training with regularization, which can result in sparsity. On the other hand, this work directly targets at optimizing the sparsity level and can thereby achieve higher sparsity degree.
>
>
> In fact our results in the ICLR workshop are also outdated. The new results, algorithm extensions, and model releases are summarized in our new Arxiv report
>
> https://arxiv.org/pdf/1804.03294.pdf

---

### Decision · Program_Chairs · 2018-03-20
**ICLR 2018 Workshop Acceptance Decision**

**Decision:**

Accept

**Comment:**

Congratulations, your paper was accepted to the ICLR workshop.